# SlimMoE: Structured Compression of Large MoE Models via Expert Slimming and Distillation

**Zichong Li**[1][†][*] **Chen Liang**[2][†] **Zixuan Zhang**[1]**, Ilgee Hong**[1]**, Young Jin Kim**[2]**,**
**Weizhu Chen**[2]**, Tuo Zhao**[1]
[1]Georgia Institute of Technology [2]Microsoft

## Abstract

The Mixture of Experts (MoE) architecture has emerged as a powerful paradigm for scaling large language models (LLMs) while maintaining inference efficiency. However, their substantial memory requirements make them prohibitively expensive to fine-tune or deploy in resource-constrained environments. To address this challenge, we propose *SlimMoE*, a multi-stage compression framework that transforms large MoE models into significantly smaller and more efficient variants without the cost of training from scratch. Our method systematically reduces parameter counts by slimming experts and transferring knowledge through intermediate stages, effectively mitigating the performance degradation typical of one-shot pruning. Using SlimMoE, we compress Phi-3.5-MoE (41.9B total / 6.6B activated parameters) into two smaller models: Phi-mini-MoE (7.6B total / 2.4B activated) and Phi-tiny-MoE (3.8B total / 1.1B activated), using only 400B tokens – less than 10% of the original training data. These models can be fine-tuned on a single GPU (A100 for Phi-mini-MoE, A6000 for Phi-tiny-MoE), making them well suited for academic and resource-limited settings. Our experiments show that the compressed models outperform others of similar size and remain competitive with larger models. For example, Phi-mini-MoE matches or exceeds the performance of Phi-3-mini while using only two-thirds of the activated parameters and achieves comparable MMLU scores to LLaMA 3.1 8B with significantly lower latency. These results highlight that structured pruning combined with multi-stage distillation is an effective strategy for building high-quality, compact MoE models, enabling broader adoption of MoE architectures across diverse computational environments. We release our models at https://huggingface.co/microsoft/Phi-mini-MoE-instruct and https://huggingface.co/microsoft/Phi-tiny-MoE-instruct.

## 1 Introduction

Large language models (LLMs) have demonstrated remarkable capabilities across various domains (Dubey et al., 2024; DeepSeek-AI et al., 2024b). The Mixture of Experts (MoE) architecture has emerged as a particularly effective approach in open-source LLMs (DeepSeek-AI et al., 2024b; Abdin et al., 2024; Jiang et al., 2024; Team, 2024), offering superior performance compared to dense models while maintaining greater inference efficiency through sparse parameter activation.

Despite these advantages, state-of-the-art MoE models developed by industry leaders typically contain an enormous number of parameters and rely on abundant computational resources. Examples include Phi-3.5-MoE (Abdin et al., 2024) and DeepSeek-V3 (DeepSeek-AI et al., 2024b), which are prohibitively large and extremely costly to train and deploy. These high costs limit their practical use in resource-constrained environments such as academic

---

[*]Work is done during internship at Microsoft.
[†]Correspondence to zli911@gatech.edu and chenliang1@microsoft.com.

research, making it difficult for researchers to fine-tune such models and highlighting the need for smaller, more efficient MoE alternatives.

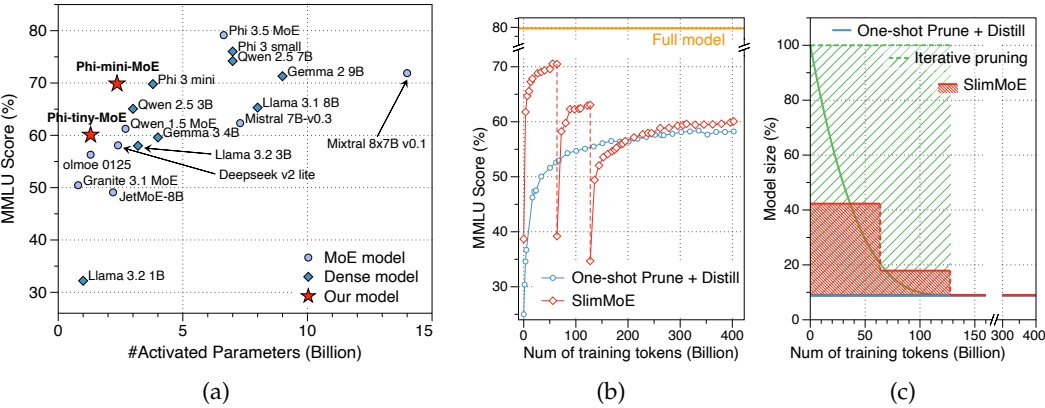

Figure 1: (a) Comparison of MMLU scores with recent open-sourced MoE and dense LLMs. (b) Comparison of MMLU scores between one-shot approach and SlimMoE multi-stage approach on Phi-tiny-MoE. (c) Illustration of model size change throughout the compression process. Shaded area indicate computation overhead of approaches over one-shot approach.

Training smaller MoE models from scratch is prohibitively expensive in terms of time, data, and computational resources (Team, 2024; Muennighoff et al., 2024). Given the availability of well-trained large MoE models, this work aims to obtain smaller, more efficient MoE models by compressing larger ones while preserving performance. This approach leverages existing models without incurring full-scale training costs. Specifically, we use the pre-trained Phi-3.5-MoE model (Abdin et al., 2024) as our starting point and scale it down to two target sizes: 7.6B total parameters (2.4B active) and 3.8B total parameters (1.1B active). These sizes are strategically chosen to enable fine-tuning on widely available hardware, e.g., the 7.6B model can be fine-tuned on a single A100 80GB GPU using memory-efficient optimizers such as 8-bit Adam (Dettmers et al., 2021) or Muon (Jordan et al., 2024), while the 3.8B model can be fine-tuned on an A6000 48GB GPU.

Among various compression methods, we focus on *structured pruning* (Ma et al., 2023; Xia et al., 2024; Muralidharan et al., 2024; Xia et al., 2022; Liang et al., 2023), which reduces parameter counts by systematically removing blocks of weights. However, compressing MoE models at high pruning ratios while preserving performance remains a significant challenge. A common strategy is to apply *one-shot pruning* to shrink the model to the target size in a single step, followed by *knowledge distillation* to recover performance (Muralidharan et al., 2024). Yet, pruning a large number of parameters at once can result in substantial performance degradation (Figure 1(b)), especially when entire architectural components, such as experts, are removed (Muzio et al., 2024; Chen et al., 2022; Chowdhury et al., 2024). Such severe degradation may hinder the effectiveness of distillation in recovering the original performance (Cho & Hariharan, 2019; Mirzadeh et al., 2020). An existing approach to address this challenge is iterative pruning (Liang et al., 2023), which gradually removes parameters during knowledge distillation to avoid a large performance drop at once and facilitate progressive recovery over time. However, iterative pruning is expensive, as it uses masks to simulate pruning and requires loading the full model throughout the process (Figure 1 (c)), making it impractical for compressing LLMs.

To address these challenges, we propose *SlimMoE*, a multi-stage expert slimming and distillation framework that mitigates the performance degradation of one-shot pruning with only modest additional computation. *SlimMoE* compresses MoE models at high ratios while preserving performance, using less than 10% of the original training data. It introduces two key design components: (1) Instead of pruning entire experts, SlimMoE retains all experts and prunes only redundant neurons within each one, preserving expert-specific knowledge critical to model performance (Section 4.5). (2) Rather than pruning directly to the target size, SlimMoE first applies one-shot pruning to an intermediate size to avoid drastic performance degradation, followed by knowledge distillation to restore accuracy

(Figure 1(b)). This prune-and-distill process is repeated until the target size is reached, with extended distillation applied at the final stage. As intermediate models retain sufficient capacity for knowledge transfer and recover quickly, this multi-stage approach ensures a smoother and more stable compression trajectory. Compared to iterative pruning, SlimMoE avoids pruning masks and full-model loading, resulting in significantly lower computational overhead (see Figure 1(c) for illustration and Section 4.2 for details).

Using our proposed method, we introduce Phi-mini-MoE (7.6B total / 2.4B active parameters) and Phi-tiny-MoE (3.8B total / 1.1B active parameters), which are compressed from Phi-3.5-MoE (41.9B total/6.6B activated parameters) with 400B tokens of continual pre-training[1]. As shown in Figure 1(a), both models substantially outperform dense and MoE baselines with similar active parameter counts and achieve competitive performance compared to models with higher inference costs. For example, Phi-mini-MoE reaches similar MMLU scores to Phi-3-mini and LLaMA 3.1 8B, while Phi-tiny-MoE outperforms LLaMA 3.2 3B and matches the performance of Qwen 1.5 MoE (See Figure 4 for comparison on inference costs). In summary, our contributions are as follows:

• We propose SlimMoE, a multi-stage expert slimming and distillation framework that enables high-ratio compression of MoE models while preserving competitive performance.

• We release two compact MoE models, Phi-mini-MoE and Phi-tiny-MoE, which achieve competitive performance among models of comparable size.

• We conduct extensive ablation studies to validate the effectiveness of SlimMoE and show that MoE architectures can be more robust to pruning than their dense counterparts.

## 2 Background

### 2.1 Mixture of Experts Models

The Mixture of Experts (MoE) architecture enhances the standard Transformer by replacing its feed-forward network (FFN) layers with MoE layers, where each input token activates only a subset of experts. This sparse activation enables the model to scale in capacity without a proportional increase in computational cost (Fedus et al., 2022; Lepikhin et al., 2020; Shazeer et al., 2017; Jiang et al., 2024; Abdin et al., 2024; Liu et al., 2024b).

An MoE layer consists of $n_{\text{expert}}$ expert networks and a router. Given an input $\mathbf{x} \in \mathbb{R}^{d_{\text{model}}}$, each expert typically adopts a Gated Linear Unit (GLU) structure (Shazeer, 2020):

$$\text{Expert}_e(\mathbf{x}) = (\text{Act}(\mathbf{x}W_{1e}^E) \odot \mathbf{x}W_{2e}^E)W_{3e}^E,$$

where $W_{1e}^E, W_{2e}^E \in \mathbb{R}^{d_{\text{model}} \times d_{\text{expert}}}$, $W_{3e}^E \in \mathbb{R}^{d_{\text{expert}} \times d_{\text{model}}}$, $e \in \{1, \ldots, n_{\text{expert}}\}$, Act$(\cdot)$ is an activation function (e.g., GELU), and $\odot$ denotes element-wise multiplication.

The router is parameterized by $W^G \in \mathbb{R}^{d_{\text{model}} \times n_{\text{expert}}}$ and produces $n_{\text{expert}}$-dimensional scores:

$$G(\mathbf{x}) = \text{Gating}(\text{TopK}(\mathbf{x}W^G)),$$

where $\text{TopK}(\mathbf{x}W^G)_i = (\mathbf{x}W^G)_i$ if $i$ is among the top-$k$ highest routing logits, and $-\infty$ otherwise. Gating$(\cdot)$ is a gating function (e.g., softmax). The output of the MoE layer is:

$$\text{MoE}(\mathbf{x}) = \sum_{e=1}^{n_{\text{expert}}} G(\mathbf{x})_e \cdot \text{Expert}_e(\mathbf{x}).$$

In this paper, we focus on Phi-3.5-MoE (Abdin et al., 2024), which sets $n_{\text{expert}} = 16$, selects the top-2 experts, and uses the SparseMixer-v2 routing mechanism (Liu et al., 2023c;b; 2024b). SparseMixer-v2 enables gradient flow through routing by replacing hard TopK with stochastic expert selection and applying a third-order approximation to the routing gradient, improving training stability and scalability.

---

[1]The released models (instruct version) are further post-trained with supervised fine-tuning (SFT) and direct preference optimization (DPO) for instruction following and preference alignment.

The self-attention mechanism operates on an input sequence $\mathbf{X} \in \mathbb{R}^{N \times d_{\text{model}}}$, where $N$ is the sequence length (Vaswani et al., 2017). Multi-head self-attention is computed as:

$$\text{MHA}(\mathbf{X}) = \text{Concat}(\text{head}_1, \ldots, \text{head}_H)W^O, \qquad \text{head}_h = \text{Attn}(\mathbf{X}W_h^Q, \mathbf{X}W_h^K, \mathbf{X}W_h^V),$$

where $H$ is the number of heads; $W_h^Q, W_h^K, W_h^V \in \mathbb{R}^{d_{\text{model}} \times d_{\text{head}}}$ are the query, key, and value projection matrices for head $h$; and $W^O \in \mathbb{R}^{H \cdot d_{\text{head}} \times d_{\text{model}}}$ is the output projection matrix.

Further, grouped-query attention (GQA, Ainslie et al. (2023)) offers an efficient alternative to full multi-head attention. It partitions the $H$ query heads into multiple groups and lets every head in a group share the same key and value projection.

## 2.2 Weight Pruning

Weight pruning is an effective compression technique that removes parameters from a model based on predefined importance criteria (Han et al., 2015b;a; Louizos et al., 2017). Commonly used metrics are *sensitivity* and its variants (Molchanov et al., 2016; 2019; Sanh et al., 2020; Zhang et al., 2022), which estimate the impact of removing individual parameters. Let $W$ denote the model weights and $L(\mathbf{X}; W)$ the loss function. The sensitivity score for weight $W_{i,j}$ is computed as:

$$s_{i,j} = \left| \frac{\partial L}{\partial W_{i,j}} \odot W_{i,j} \right|,$$

which approximates the change in loss when setting $W_{i,j}$ to zero, using a first-order Taylor expansion of $L$ around $W$. Weights with low sensitivity are considered redundant.

Pruning methods are typically categorized into *unstructured* and *structured* approaches. Unstructured pruning removes individual weights, resulting in sparse weight matrices. While this often causes minimal performance degradation, it does not yield practical speedup without specialized hardware support and tensor structure (Fang et al., 2024). Structured pruning, on the other hand, removes entire architectural components (e.g., neurons, attention heads, or experts), which enables real efficiency gains on standard hardware but may cause greater disruption to model performance and has been widely adopted for compressing LLMs in recent years (Ma et al., 2023; Xia et al., 2024; Muralidharan et al., 2024; Xia et al., 2022; Liang et al., 2023).

## 2.3 Knowledge Distillation

Knowledge Distillation (KD) is a widely used compression technique that trains a smaller student model to mimic the behavior of a larger teacher model by minimizing the discrepancy between their output distributions (Hinton et al., 2015; Muralidharan et al., 2024). The objective is typically formulated as the Kullback–Leibler (KL) divergence:

$$L_{\text{KD}}(\mathbf{X}; W) = \frac{1}{N} \sum_{n=1}^{N} \text{KL}\left( p_{\text{teacher}}^n(\mathbf{X}) \parallel p_W^n(\mathbf{X}) \right), \tag{1}$$

where $W$ denotes the student's parameters, $p_{\text{teacher}}^n(\mathbf{X})$ and $p_W^n(\mathbf{X})$ are the teacher and student next-token distributions for the $n$-th token in input $\mathbf{X}$.

To improve the efficiency and robustness of knowledge distillation, Peng et al. (2024) proposed to minimize the discrepancy over the $k$ tokens with the highest predicted probabilities instead of the full vocabulary distribution, reducing computational requirements and mitigating noise from low-probability tokens.

## 3 Method

SlimMoE employs a multi-stage framework that progressively reduces model size by alternating expert and attention pruning with knowledge distillation. We begin by describing the target MoE architecture, followed by our pruning and distillation strategies.

### 3.1 Target Architecture

**Expert Slimming**. Since MoE layers account for over 95% of the model's parameters, we focus on pruning these layers. Specifically, we reduce the expert dimension $d_{\text{expert}}$ by pruning redundant neurons within each expert network, i.e., $\{W_{1e}^E, W_{2e}^E, W_{3e}^E\}_{e=1}^{n_{\text{expert}}}$ in Eq. 2.1, while keeping the number of experts fixed. As experts often serve specialized roles for subsets of tokens, this approach better preserves expert functionality and results in smaller accuracy degradation than pruning entire experts (Section 4.5). We apply uniform slimming across all experts to maintain equal expert size for architectural consistency and deployment.

**Attention Pruning**. We also prune attention layers as they begin to dominate the parameter count and inference time at smaller scales. Since Phi-3.5-MoE adopts GQA that ties four query heads to a shared key/value projection, we prune at the granularity of entire GQA groups, eliminating both the shared KV pair and its four associated queries.

**Target Architecture**. We reduce the expert dimension of Phi-3.5-MoE (41.9B total / 6.6B activated parameters) to 15% of its original size to yield Phi-mini-MoE (7.6B total / 2.4B activated parameters). To obtain Phi-tiny-MoE (3.8B total / 1.1B activated parameters), we further reduce the expert dimension to 7% and prune 50% of the GQA groups. Table 6 in the Appendix summarizes the detailed architecture configurations.

### 3.2 Multi-stage Pruning and Distillation

We prune and distill the model over $T$ stages, with each stage producing a smaller model than the previous one. At each stage, we apply one-shot pruning to reach an intermediate size, followed by distillation from the original full model.

**One-shot Pruning**. We use the top-8 logits distillation loss to compute the sensitivity score for each parameter $W_{i,j}$:

$$s_{i,j} = \left| \frac{\partial L_{\text{KD}}^{\text{MoE}}(\mathbf{X}; W)}{\partial W_{i,j}} \odot W_{i,j} \right|, \tag{2}$$

where the loss is defined as:

$$L_{\text{KD}}^{\text{MoE}}(\mathbf{X}; W) = \frac{1}{N} \sum_{n=1}^{N} \text{KL}\left( p_{\text{teacher, top-8}}^n(\mathbf{X}) \parallel p_W^n(\mathbf{X}) \right) + \text{Aux}(W). \tag{3}$$

Here, $p_{\text{teacher, top-8}}^n(\mathbf{X})$ is the teacher's next-token prediction probability distribution for the $n$-th token with all but the top-8 probabilities masked to 0, and $p_W^n(x)$ is the student distribution as defined in Eq. 1. $\text{Aux}(\cdot)$ is the auxiliary load-balancing loss (Liu et al., 2024b).

We compute the sensitivity score using the knowledge distillation loss as it captures the discrepancy between the student and teacher. This allows us to identify redundant neurons that have negligible impact on the gap, yielding better pruning performance than alternative loss metrics (Section 4.5).

For the $e$-th expert network, we first compute the sensitivity score for each parameter in the down-projection matrix $W_{3e}^E$ of the GLU. We then aggregate these into a neuron-level score by taking the $\ell_2$-norm across each row:

$$s_i = \sqrt{\sum_j s_{i,j}^2}. \tag{4}$$

Neurons with the lowest scores are considered least important. We prune these by removing the corresponding rows and columns in $W_{1e}^E$, $W_{2e}^E$, and $W_{3e}^E$. For attention pruning, we apply Eq. 4 to the output projection matrix $W^O$ (Eq. 2.1) and average the scores of all neurons within heads in a GQA group to obtain the score for each group. We uniformly sample 16K training examples as calibration data to compute sensitivity scores.

**Distillation**. After pruning, we distill the model using the original full model as the teacher to recover performance. The student model is optimized using a gradient-based method (Loshchilov & Hutter, 2017):

$$W \leftarrow W - \eta \nabla_W L_{\text{KD}}^{\text{MoE}}(x; W), \qquad (5)$$

where $\eta$ is the learning rate. To reduce computational overhead, we apply early stopping once performance improvements begin to plateau, rather than training each intermediate model to full convergence. In practice, distillation at intermediate stages consumes only 30-35% of the total training steps (Figure 2).

**Multi-stage Schedule**. We use two stages ($T$=2) for Phi-mini-MoE, and for the more aggressively compressed Phi-tiny-MoE, we use three stages ($T$=3).

To ensure balanced pruning across stages, we follow a geometric compression schedule. Given a target overall compression ratio $\alpha$, we reduce the model size at each stage by a factor of approximately $\alpha^{1/T}$. The full architectural specifications for each intermediate model are provided in Appendix A.3. This progressive strategy allows each intermediate model to retain sufficient capacity for effective knowledge transfer, resulting in a smoother and more stable transition to the final compact model.

## 4 Experiments

We compress the pre-trained Phi-3.5-MoE model (Abdin et al., 2024; Liu et al., 2024b) to obtain Phi-mini-MoE and Phi-tiny-MoE base models through continual pre-training. To better evaluate on downstream tasks, we further post-train the two base models by supervised fine-tuning (SFT) to enhance the models' instruction-following capabilities, followed by Direct Preference Optimization (DPO, Rafailov et al. (2023)) to steer the model away from unwanted behavior.

**Data and Training.** For continual pre-training, we perform multi-stage distillation using a 400B-token subset of the Phi-3.5-MoE (Abdin et al., 2024; Liu et al., 2024b) pre-training corpus, leveraging the top-8 logits predicted by the Phi-3.5-MoE teacher. For post-training, we apply SFT and DPO using the GRIN-MoE (Liu et al., 2024b) post-training corpus[2]. During SFT, we adopt a top-8 logits distillation objective using the post-trained GRIN-MoE as the teacher. The training hyperparameter configurations are provided in the Appendix A.5.

**Evaluation.** We evaluate our models against other similarly-sized models, including both MoE and dense architectures. For MoE models, we include Qwen 1.5 MoE (Team, 2024), DeepSeek V2 Lite (DeepSeek-AI et al., 2024a), OL-MoE (Muennighoff et al., 2024), and Granite 3.0 (Granite Team, 2024). For dense models, we include the Phi-3 series (Abdin et al., 2024), Llama 3 series (Dubey et al., 2024), Qwen 2.5 series (Yang et al., 2024a), and Gemma 3 series (Team et al., 2025).

We evaluate these models across a diverse set of downstream tasks. For commonsense and knowledge assessment, we employ MMLU (Hendrycks et al., 2021), MMLU-pro (Wang et al., 2024), Bigbench-Hard (Suzgun et al., 2023), Arc-Challenge (Clark et al., 2018), HellaSwag (Zellers et al., 2019), OpenbookQA (Mihaylov et al., 2018), PIQA (Bisk et al., 2020), BoolQ (Clark et al., 2019), and Winograde (Sakaguchi et al., 2020). To evaluate coding capabilities, we include HumanEval (Chen et al., 2021) and MBPP (Austin et al., 2021), while for reasoning and mathematical tasks, we utilize GSM8K (Cobbe et al., 2021) and GPQA (Rein et al., 2023). Additionally, we report MT-bench (Zheng et al., 2023) scores to assess general instruction-following abilities. We provide the detailed settings for evaluations in Appendix A.6.

### 4.1 Performance of Compressed Models

Table 1 shows the evaluation results comparing the post-trained Phi-mini-MoE and Phi-tiny-MoE against other MoE and dense models of comparable sizes. More benchmark results are shown in Table 4, and the performance of the pretrained models is reported in Table 5.

---

[2]We follow GRIN-MoE's post-training setup for its simplicity, as Phi-3.5-MoE employs more complexity to build up long-context and multilingual capabilities.

Table 1: Comparison of Phi-mini-MoE and Phi-tiny-MoE against other models.

| Model | # Total param | # Act. param | MMLU | MMLU pro | BBH | Arc-C (chat) | Human-eval | GSM8K | MT-bench |
|---|---|---|---|---|---|---|---|---|---|
| **Mixture-of-Experts (MoE) Models** | | | | | | | | | |
| Phi-3.5-MoE | 42B | 6.6B | 78.36 | 59.38 | 63.93 | 91.38 | 81.70 | 87.87 | 8.34 |
| Qwen 1.5 MoE | 14B | 2.7B | 60.73 | 26.49 | 42.65 | 67.24 | 46.30 | 53.07 | 6.55 |
| DeepSeek V2 Lite | 16B | 2.4B | 56.69 | 17.89 | 36.30 | 61.09 | 54.40 | 63.23 | 6.82 |
| OL-MoE | 6.9B | 1.3B | 54.27 | 20.87 | 38.00 | 55.63 | 37.80 | 71.49 | 6.60 |
| Granite 3.0 MoE | 3.4B | 0.8B | 50.06 | 4.82 | 39.65 | 56.06 | 51.80 | 60.12 | 6.91 |
| **Dense Models** | | | | | | | | | |
| LLaMA 3.1 8B | 8B | 8B | 68.71 | 45.28 | 50.86 | 82.42 | 69.50 | 84.84 | 8.03 |
| Qwen 2.5 7B | 7.6B | 7.6B | 73.47 | 56.24 | 53.74 | 88.82 | 81.70 | 84.84 | 8.34 |
| Phi-3-small | 7.4B | 7.4B | 75.35 | 52.06 | 62.07 | 84.30 | 70.10 | 84.84 | 8.03 |
| Gemma 3 4B | 4.3B | 4.3B | 59.49 | 40.13 | 49.45 | 75.85 | 67.10 | 78.92 | 8.28 |
| Phi-3-mini | 3.8B | 3.8B | 69.94 | 45.65 | 54.94 | 85.58 | 72.60 | 84.61 | 7.46 |
| LLaMA 3.2 3B | 3.2B | 3.2B | 61.73 | 36.70 | 45.46 | 75.77 | 52.40 | 77.41 | 7.46 |
| Qwen 2.5 3B | 3.1B | 3.1B | 65.06 | 41.00 | 46.61 | 80.20 | 73.80 | 76.57 | 7.60 |
| Gemma 3 1B | 1B | 1B | 40.80 | 14.70 | 34.80 | 37.46 | 41.50 | 41.77 | 6.67 |
| LLaMA 3.2 1B | 1.2B | 1.2B | 46.30 | 18.67 | 35.18 | 49.91 | 35.40 | 44.96 | 5.23 |
| **Our Models** | | | | | | | | | |
| Phi-mini-MoE | 7.6B | 2.4B | 70.68 | 49.68 | 55.27 | 84.91 | 73.80 | 84.89 | 7.59 |
| Phi-tiny-MoE | 3.8B | 1.1B | 60.83 | 36.34 | 45.58 | 76.37 | 58.50 | 78.47 | 7.05 |

Our compressed models demonstrate strong performance while maintaining parameter efficiency. Phi-mini-MoE matches or outperforms Phi-3-mini across tasks with only two-thirds of its activated parameters, and achieves performance on par with Phi-3-small on ARC-Challenge, HumanEval, and GSM8K using just one-third of the activation size. Compared to other public dense models with similar total parameter counts, Phi-mini-MoE outperforms LLaMA 3.1 8B on most benchmarks and surpasses Qwen 2.5 7B on BBH, HellaSwag, and OpenBookQA, while using only one-third of their activated parameters. Among public MoE models, Phi-mini-MoE outperforms Qwen 1.5 MoE and DeepSeek V2 Lite MoE, while having less total parameters. Phi-tiny-MoE also shows strong results, outperforming OL-MoE and Granite 3.0 MoE with similar activation parameter counts and achieving performance comparable to LLaMA 3.2 3B at similar total parameter counts.

We note that the Phi-mini-MoE and Phi-tiny-MoE base models prior to post-training maintain similar relative performance compared to their respective baselines (Table 5), confirming that SlimMoE effectively preserves the model's capabilities.

## 4.2 Comparing Multi-stage with One-stage and Iterative Pruning

We compare SlimMoE with the conventional one-stage baseline (Muralidharan et al. (2024)) and iterative pruning (Liang et al. (2023) without the layerwise distillation objective) to demonstrate the effectiveness of our multi-stage design.

To ensure fair comparison, we control the total number of training tokens throughout the process to match between baselines. For iterative pruning, we set the pruning schedule to reach the target size at the same number of tokens as SlimMoE's initialization stages (stages before the final stage). As shown in Table 2, our multi-stage approach consistently outperforms the one-stage method across all evaluated benchmarks. On Phi-tiny-MoE, our approach achieves similar performance to iterative pruning despite the latter's much higher computational costs, as iterative pruning requires loading the full-size model at the pruning stage, resulting in approximately 2.4× computation time during initialization phase.

Figure 2 (a) illustrates how the MMLU scores evolve throughout the compression process for Phi-mini-MoE, while the corresponding plot for Phi-tiny-MoE is shown in Figure 1 (b). We can observe that the one-stage approach causes model collapse at the initial pruning step, whereas SlimMoE yields a smaller drop. The gradual transition through intermediate model sizes enables the knowledge distillation process to be more effective at each stage, resulting in superior overall performance in the final compressed model. While the multi-stage approach incurs additional computation during the initialization phases, it reaches the final convergence performance of the one-stage baseline earlier, resulting in reduced computation time to achieve the same performance: 0.74× for Phi-mini-MoE and 0.91× for Phi-tiny-MoE.

Table 2: Performance comparison of multi-stage, iterative and one-stage pruning approaches. Models evaluated here are pretrained version.

| Model Type | Approach | MMLU | Arc-C | WinoGrande | HellaSwag | GSM8K |
|---|---|---|---|---|---|---|
| Phi-mini-MoE | One-Stage | 68.58 | 60.84 | 70.85 | 75.93 | 74.52 |
| | Multi-Stage | **69.87** | **62.29** | **75.85** | **76.03** | **77.48** |
| Phi-tiny-MoE | One-Stage | 58.40 | 56.99 | 70.80 | 66.24 | 62.89 |
| | Iterative | 60.05 | 56.71 | **72.20** | **67.52** | 69.60 |
| | Multi-Stage | **60.08** | **57.68** | 71.90 | 67.33 | **69.90** |

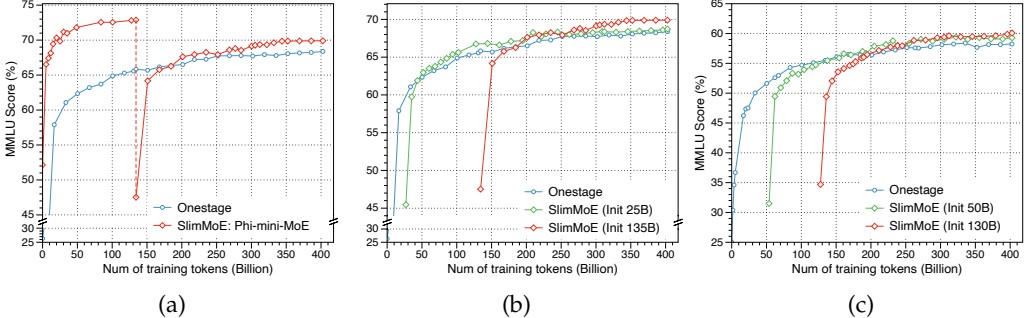

(a)               (b)               (c)

Figure 2: MMLU performance analysis across compression stages. (a) Comparison between SlimMoE and one-stage approach on Phi-mini-MoE. (b) Impact of initialization token counts on Phi-mini-MoE. (c) Impact of initialization token counts on Phi-tiny-MoE.

### 4.3 When to Stop Previous Stages

A critical question for the proposed SlimMoE framework is determining the timing to stop current stage and proceed to the next. To address this question, we experiment with different token allocations for the initialization phases (where "initialization" refers to all training conducted before the final stage). As illustrated in Figure 2 (b) and (c), we compare both shorter and longer initialization approaches. For Phi-mini-MoE, we evaluate initializations of 25B and 135B tokens, while for Phi-tiny-MoE, we test 50B and 130B tokens. We observe that longer initialization consistently leads to better final performance on both model sizes. This indicates that extended earlier stages facilitate more effective knowledge transfer to subsequent stages. In practice, we recommend to initiate the next stage when the performance improvement in the current stage becomes minimal.

### 4.4 Are MoE Models Easier to Prune than Dense Models?

In this subsection, we investigate whether MoE architecture is more robust to pruning compared to dense model by conducting one-stage prune-and-distill on both MoE and dense architectures. For a fair comparison, we choose Phi-3-medium as the dense counterpart because it achieves performance comparable to Phi-3.5-MoE, and both models adopt the same pre-training data sources. We then compress Phi-3-medium based on the compression ratio of the activated parameters of our MoE models: 35% for Phi-mini-MoE and 16% for Phi-tiny-MoE. Detailed architecture configurations can be found in Appendix A.3.

Figure 3 presents the MMLU scores for both model types across different numbers of training tokens. The results show a consistent performance advantage for pruned MoE models over their dense counterparts. The improvements become even more pronounced at the more aggressive 16% compression level. These findings suggest that MoE architectures may indeed be more amenable to pruning, potentially due to their inherent sparse activation patterns and distributed knowledge representation across experts.

### 4.5 Pruning Architecture and Criterion

SlimMoE employs top-8 logits knowledge distillation (KD) loss for computing sensitivity scores to slim all experts. To justify this design choice, we conduct comparative experiments

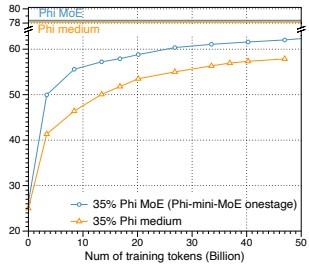 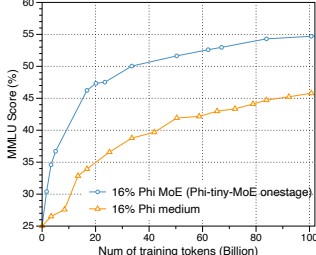

Figure 3: Performance of pruned Phi-3.5-MoE versus pruned Phi-3-medium. Left: pruned ratio 35%; Right: pruned ratio 16%.

with various pruning approaches for MoE layers. We evaluate five methods: (1) Expert slimming with top-8 KD loss (2) Expert slimming with causal language modeling loss (CLM) (3) Expert pruning based on sensitivity scores computed from KD loss (4) Expert pruning based on activation frequency (Muzio et al., 2024) (5) M-SMoE: Merging experts based on routing logits (Li et al., 2024).

Table 3: Ablation studies on pruning architecture and criterion.

| Prune Ratio | Expert Slimming (KD) | Expert Slimming (CLM) | Prune Expert (KD) | Prune Expert (freq) (Muzio et al., 2024) | M-SMoE (Li et al., 2024) |
|---|---|---|---|---|---|
| 50% | **63.76** | 59.30 | 53.41 | 48.59 | 38.54 |
| 25% | **43.25** | 37.52 | 30.38 | 31.87 | 25.50 |

As shown in Table 3, using KD loss consistently outperforms CLM loss across different pruning ratios. This improvement likely stems from the teacher model's ability to mitigate noise in the training data. Our results also show that expert slimming outperforms expert pruning, suggesting that all experts contain specialized knowledge, and slimming them better preserves this knowledge compared to removing entire experts. We include studies on expert diversity in Appendix A.8. Merging experts similarly proves ineffective, potentially due to the increased size and heterogeneous weight distributions within each expert.

## 4.6 Inference cost

We conduct inference speed profiling for our compressed models and compare them with models of similar performance or activated parameter counts in Figure 4. We collect the inference metrics using the inference benchmarking scripts implemented by DeepSpeed[3]. More details of the profiling can be found in Appendix A.7. The results demonstrate that our compressed models achieve lower latency and comparable or higher throughput across different client loads. Notably, Phi-mini-MoE and Phi-tiny-MoE maintain these computational efficiency advantages while yielding competitive performance.

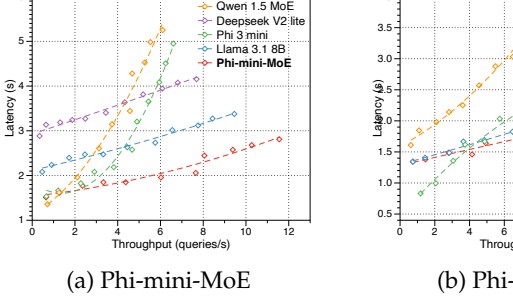

(a) Phi-mini-MoE      (b) Phi-tiny-MoE

Figure 4: Latency versus throughput comparison across models under varying client loads.

---

[3]https://github.com/deepspeedai/DeepSpeedExamples/tree/master/benchmarks/inference/mii

## 5  Discussion

**MoE Compression**. Previous MoE compression techniques can be broadly categorized into three approaches: expert-level pruning, expert merging, and unstructured expert slimming. Expert-level pruning methods (Muzio et al., 2024; Lu et al., 2024) identify and remove redundant experts. Muzio et al. (2024) uses router logits and activation count to identify redundant experts, leading to large performance drop with high compression ratio (Section 4.5). Lu et al. (2024) enumerates all possible expert combinations for each layer to find the one with lowest reconstruction loss, which can becomes computationally prohibitive for large models with many experts. Expert merging techniques (Liu et al., 2024a; Li et al., 2024) attempt to preserve knowledge by combining experts rather than removing them. Li et al. (2024) aligns neurons across experts and then merges them based on router information, but does not work well in our settings. Liu et al. (2024a) focuses on task-specific settings and employs evolutionary search to identify merging weights. This approach requires hundreds of iterations to converge, making it substantially more expensive than our pruning method. Additionally, determining an appropriate metric for task-agnostic setting is nontrivial. He et al. (2025) and Xie et al. (2024) study unstructured expert slimming, which cannot leads to efficiency without specified hardware. Some other works consider techniques besides pruning for compressing MoE, operating on different dimensions and are complementary to SlimMoE. Yang et al. (2024b) first employ Layer-wise expert-level non-uniform pruning, then apply Singular Value Decomposition to further compress the remaining experts. Kim et al. (2023) utilize quantization to MoE models, reducing memory through lower numerical precision rather than architectural changes.

**Dense Model Compression**. Dense model compression has been extensively studied in recent literature (Xia et al., 2024; Muralidharan et al., 2024; Men et al., 2024). For instance, ShortGPT (Men et al., 2024) and LaCo (Yang et al., 2024c) propose to remove entire layers. Other works (Xia et al., 2024; Ashkboos et al., 2024; Ma et al., 2023) introduce various strategies for compressing width dimensions. Minitron (Muralidharan et al., 2024) represents a notable work that shares similarities with our approach. They focus on dense models and employ one-shot pruning and distillation method on Nemotron 4 15B (Parmar et al., 2024) to produce Minitron 8B, followed by further pruning to 4B, which is a two-stage compression pipeline. However, their comparison with single-stage approaches does not account for the tokens used in the initial stage. In contrast, SlimMoE introduces a principled multi-stage approach for MoE models. We systematically studies the multi-stage approach by experimenting with different stage lengths and numbers, demonstrating that multi-stage outperforms one-stage approaches under the same token budget.

**Computational Cost for the Multi-stage Approach**. While our experiments on Phi-3.5-MoE requires 400B tokens for performance recovery, the computational overhead remains manageable due to our strategic token allocation. Since most tokens are trained in the final stage when the model is heavily pruned, the overall computational cost is substantially reduced compared to training target-size models from scratch, which would require the full 4T tokens. SlimMoE also demonstrates robustness to varying token budgets, enabling users to adjust computational resources based on their constraints. To illustrate this flexibility, we provide additional results under a reduced training budget of 90B tokens in Appendix A.4.

## 6  Conclusion

In this paper, we presented SlimMoE, a multi-stage framework for compressing large MoE models by high ratios. Using SlimMoE, we compressed Phi 3.5-MoE to create Phi-mini-MoE and Phi-tiny-MoE using only 10% of the original pretraining data. These models significantly outperform both MoE and dense models with similar activated parameter counts. Our experiments demonstrate the effectiveness of the multi-stage approach and highlight the importance of preserving knowledge across experts through expert slimming. Notably, while our evaluation focuses on Phi-series models, SlimMoE is architecturally agnostic, making it broadly applicable to other MoE architectures. To the best of our knowledge, this is the first work to prune large MoE models at such high ratios ($<$20% of original parameters) while achieving state-of-the-art performance.

## Acknowledgment

We would like to thank Shuohang Wang for the helpful discussion on distillation.

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

# A  Appendix

## A.1  Results on extended tasks.

Table 4: Extended evaluation on commonsense reasoning, QA, and code generation tasks.

| Model | # Total param | # Act. param | Winograde | Hellaswag | GPQA | PIQA | OpenbookQA | BoolQ | MBPP |
|---|---|---|---|---|---|---|---|---|---|
| **Mixture-of-Experts (MoE) Models** | | | | | | | | | |
| Phi 3.5-MoE | 42B | 6.6B | 78.61 | 82.31 | 39.90 | 81.72 | 63.20 | 88.93 | 73.30 |
| Qwen 1.5 MoE | 14B | 2.7B | 70.40 | 79.40 | 27.27 | 80.25 | 49.40 | 81.77 | 42.90 |
| DeepSeek V2 Lite | 16B | 2.4B | 70.40 | 74.81 | 25.76 | 79.60 | 46.20 | 83.30 | 59.50 |
| OL-MoE | 6.9B | 1.3B | 66.54 | 75.44 | 24.75 | 80.79 | 48.00 | 78.78 | 36.50 |
| Granite 3.0 MoE | 3.4B | 0.8B | 67.17 | 70.64 | 27.78 | 78.45 | 44.40 | 78.99 | 52.90 |
| **Dense Models** | | | | | | | | | |
| LLaMA 3.1 8B | 8B | 8B | 75.61 | 78.27 | 28.79 | 81.28 | 52.40 | 85.20 | 67.70 |
| Qwen 2.5 7B | 7.6B | 7.6B | 75.90 | 72.25 | 34.34 | 72.20 | 50.00 | 86.85 | 79.40 |
| Phi 3 small | 7.4B | 7.4B | 73.38 | 81.97 | 30.81 | 81.72 | 57.20 | 87.13 | 72.80 |
| Gemma 3 4B | 4.3B | 4.3B | 65.67 | 55.80 | 27.27 | 73.45 | 49.60 | 80.03 | 78.00 |
| Phi 3 mini | 3.8B | 3.8B | 76.95 | 76.48 | 26.77 | 81.63 | 53.80 | 85.87 | 72.50 |
| LLaMA 3.2 3B | 3.2B | 3.2B | 67.72 | 70.40 | 22.22 | 78.45 | 43.20 | 77.98 | 63.00 |
| Qwen 2.5 3B | 3.1B | 3.1B | 71.10 | 70.37 | 23.74 | 74.54 | 46.60 | 83.59 | 72.50 |
| Gemma 3 1B | 1B | 1B | 55.80 | 47.66 | 22.73 | 71.98 | 40.20 | 70.86 | 57.70 |
| LLaMA 3.2 1B | 1.2B | 1.2B | 62.04 | 61.0 | 21.21 | 75.35 | 36.0 | 57.37 | 33.10 |
| **Our Models** | | | | | | | | | |
| Phi-mini-MoE | 7.6B | 2.4B | 75.45 | 74.76 | 27.78 | 81.77 | 51.20 | 85.02 | 69.60 |
| Phi-tiny-MoE | 3.8B | 1.1B | 70.09 | 67.44 | 29.29 | 79.16 | 48.00 | 81.07 | 53.70 |

## A.2  Results of pretrained models

Results of pretrained models are presented in Table 5.

Table 5: Performance comparison of pretrained models on various benchmark tasks. * We report the performance of the instruct model for Phi 3.5-MoE.

| Model (pretrained) | # Total param | # Act. param | MMLU | Winograde | Arc-C | Hellaswag | BoolQ |
|---|---|---|---|---|---|---|---|
| **Mixture-of-Experts (MoE) Models** | | | | | | | |
| Phi 3.5-MoE* | 42B | 6.6B | 78.36 | 78.61 | 65.52 | 82.31 | 88.93 |
| Qwen 1.5 MoE | 14B | 2.7B | 61.24 | 71.82 | 56.06 | 79.17 | 80.79 |
| DeepSeek V2 Lite | 16B | 2.4B | 58.09 | 75.69 | 56.31 | 79.79 | 79.34 |
| OL-MoE | 6.9B | 1.3B | 54.96 | 72.45 | 58.53 | 79.12 | 77.86 |
| Granite 3.0 MoE | 3.4B | 0.8B | 48.44 | 67.8 | 49.57 | 73.15 | 75.38 |
| **Dense Models** | | | | | | | |
| LLaMA 3.1 8B | 8B | 8B | 65.22 | 77.03 | 57.51 | 80.89 | 82.69 |
| Qwen 2.5 7B | 7.6B | 7.6B | 74.20 | 75.90 | 63.31 | 79.46 | 87.77 |
| Gemma 3 4B | 4.3B | 4.3B | 59.51 | 72.53 | 58.36 | 77.23 | 80.03 |
| LLaMA 3.2 3B | 3.2B | 3.2B | 56.07 | 72.14 | 48.38 | 75.46 | 73.30 |
| Qwen 2.5 3B | 3.1B | 3.1B | 65.55 | 71.35 | 57.41 | 74.49 | 83.94 |
| Gemma 3 1B | 1B | 1B | 26.26 | 61.01 | 39.68 | 62.38 | 64.77 |
| LLaMA 3.2 1B | 1.2B | 1.2B | 31.00 | 62.03 | 38.30 | 65.70 | 65.50 |
| **Our Models** | | | | | | | |
| Phi-mini-MoE | 7.6B | 2.4B | 69.87 | 75.85 | 62.29 | 76.03 | 85.14 |
| Phi-tiny-MoE | 3.8B | 1.1B | 60.08 | 71.90 | 57.68 | 67.33 | 80.97 |

## A.3  Architecture of intermediate sizes and dense model

Architecture configurations for Phi-mini-MoE and Phi-tiny-MoE are shown in Table 6. Architecture configurations of intermediate sizes for Phi-mini-MoE and Phi-tiny-MoE are shown in Table 7. Architecture configurations of pruned Phi 3 medium are shown in Table 8.

Table 6: Model configuration details and parameter counts of Phi-mini-MoE and Phi-tiny-MoE.

| Model | $d_{\text{model}}$ | $n_{\text{head}}$ (q/kv) | $d_{\text{expert}}$ | $n_{\text{layer}}$ | $n_{\text{expert}}$ | top-k | # Total Param | # Act. Param |
|---|---|---|---|---|---|---|---|---|
| Phi-3.5-MoE | 4096 | 32/8 | 6400 | 32 | 16 | 2 | 41.9B | 6.6B |
| Phi-mini-MoE | 4096 | 32/8 | 960 | 32 | 16 | 2 | 7.6B | 2.4B |
| Phi-tiny-MoE | 4096 | 16/4 | 448 | 32 | 16 | 2 | 3.8B | 1.1B |

Table 7: Model configuration details and parameters counts of intermediate sizes.

| Model | $d_{\text{model}}$ | $n_{\text{head}}$ | $d_{\text{expert}}$ | $n_{\text{layer}}$ | $n_{\text{expert}}$ | top-k | # Total Param | # Act. Param |
|---|---|---|---|---|---|---|---|---|
| Phi 3.5-MoE / GRIN-MoE | 4096 | 32/8 | 6400 | 32 | 16 | 2 | 41.9B | 6.6B |
| Phi-mini-MoE-stage1 | 4096 | 32/8 | 2240 | 32 | 16 | 2 | 15.7B | 3.4B |
| Phi-mini-MoE (stage2) | 4096 | 32/8 | 960 | 32 | 16 | 2 | 7.6B | 2.4B |
| Phi-tiny-MoE-stage1 | 4096 | 24/6 | 2624 | 32 | 16 | 2 | 17.8B | 3.3B |
| Phi-tiny-MoE-stage2 | 4096 | 20/5 | 1024 | 32 | 16 | 2 | 7.6B | 1.9B |
| Phi-tiny-MoE (stage3) | 4096 | 16/4 | 448 | 32 | 16 | 2 | 3.8B | 1.1B |

## A.4 Results with lower training budgets.

We evaluated SlimMoE's effectiveness under reduced training budgets. We conducted experiments on Phi-mini-MoE's size using only 90B total tokens, with 25B tokens allocated to the first stage. Table 9 presents the performance comparison between our multi-stage approach and the one-stage baseline. The results demonstrate that even with a 4.4× reduction in training tokens, our multi-stage framework consistently outperforms the one-stage approach, confirming that SlimMoE's advantages persist even under resource constraints.

## A.5 Training Details

**Training Hyperparameters**. We use different hyperparameter settings during the compression process and subsequent SFT distillation. During compression, we use the AdamW optimizer with a learning rate of 1e-4 and weight decay of 0.01. We apply cosine learning rate decay with 100 warmup steps. For SlimMoE, we apply these warmup steps at the beginning of each stage. We use a batch size of 4096 and a maximum sequence length of 4096. For SFT distillation, we maintain the same optimizer but reduce the learning rate to 2e-6 and weight decay to 1e-4. We use a batch size of 1536. All training was conducted on 64 A100 GPUs.

## A.6 Evaluation Details

We use lm-evaluation-harness (Gao et al., 2024) for evaluations on all tasks except MT-bench, Humaneval and MBPP, where we follows original implementation of MT-bench and use evalplus base mode (Liu et al., 2023a) for coding tasks. We use different evaluation settings for pretrained and instruct model. For pretrained models, we use 5 shots for all tasks. For instruct models, we use 5 shots for MMLU, MMLU pro, Winograde, Hellaswag and PIQA, 0 shot for Arc-challenge with chat prompt, 0 shot cot for GPQA, 2 shots for BoolQ, 10 shots for openbookQA, 3 shots for BBH and 8 shots for GSM8K. We apply chat template when evaluating instruct models.

## A.7 Computational cost

For inference speed profiling, we use vLLM as the serving backend for all evaluated models. The number of concurrent clients is set as default (1, 2, 4, 6, 8, 12, 16, 20, 24, 28, 32). The prompt of request has a mean length of 2048 tokens and the generation length is 256 tokens.

Table 8: Model Configuration Details for Phi 3 Dense Models

| Model | $d_{\text{model}}$ | $n_{\text{head}}$ | $d_{\text{ffn}}$ | $n_{\text{layer}}$ | # Total Param |
|---|---|---|---|---|---|
| Phi 3 Dense | 5120 | 40/10 | 17920 | 40 | 14.0B |
| 35% Phi 3 Dense | 5120 | 40/10 | 3297 | 40 | 5.0B |
| 16% Phi 3 Dense | 5120 | 20/5 | 1098 | 40 | 1.9B |

Table 9: Performance comparison under reduced training budget (90B tokens total)

| Method | MMLU | Winograde | Arc-C | HellaSwag |
|---|---|---|---|---|
| One-stage | 64.11 | 72.34 | 59.13 | 73.23 |
| Multi-stage | 65.59 | 72.74 | 62.54 | 73.38 |

For each client load, we record both throughput total latency. We also conduct memory profiling during finetuning Phi-mini-MoE and Phi-tiny-MoE in Table 10.

Table 10: Memory Profiling Results

| Model | Optimizer | Peak Allocated Memory (MB) | Peak Reserved (MB) |
|---|---|---|---|
| Phi-mini-MoE | 8-bit Adam | 59204.31 | 68756.00 |
| | Muon | 59759.51 | 61690.00 |
| Phi-tiny-MoE | 8-bit Adam | 40573.62 | 47046.00 |
| | Muon | 40556.37 | 42862.00 |

## A.8 Expert similarity

We notice that in some existing works (Lu et al., 2024; Liu et al., 2024a), the performance drop observed with expert pruning is smaller than what we have observed on Phi-3.5-MoE. We attribute this discrepancy to the differences between model families. Previous studies mainly used Mixtral as their target model, where experts exhibit high similarity, possibly because of tailored initialization. This similarity could reduces the performance impact of expert pruning. In contrast, Phi-MoE-3.5 appears to distribute knowledge more heterogeneously across experts. We present our detailed study below:

We calculated the expert similarity in Mixtral 7×8B and Phi-3.5-MoE. Figure 5 presents the distribution of maximum cosine similarities between neurons across different experts within each model.

We randomly sample a pair of experts from each model. For each neuron in the first expert, we identified the neuron in the second expert with the highest cosine similarity, then plotted the distribution of these maximum similarity values across all neurons.

The results show that Mixtral exhibits substantially higher inter-expert similarity. We also observe that in Mixtral, the neurons with highest cosine similarity tend to be positioned at corresponding indices across experts. In contrast, Phi 3.5-MoE shows lower similarity and lacks this positional correspondence pattern.

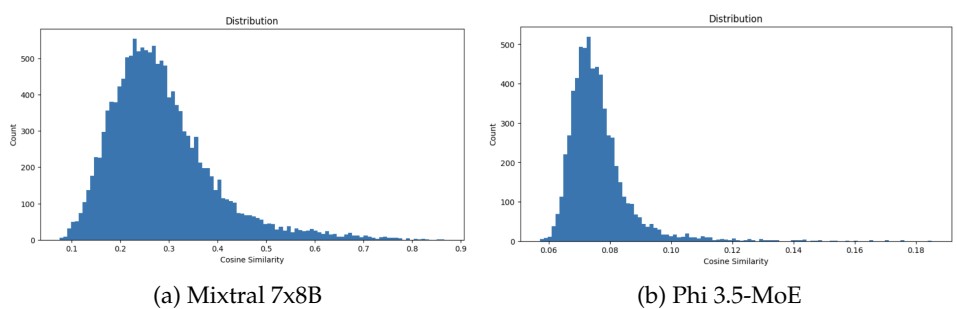

(a) Mixtral 7x8B  (b) Phi 3.5-MoE

Figure 5: Cosine similarity distribution of neurons in randomly sampled two experts

