# OpenReview forum: "SlimMoE: Structured Compression of Large MoE Models via Expert Slimming and Distillation"
_colmweb.org/COLM/2025/Conference — COLM 2025_

### Official Review · Reviewer_MkJW · 2025-05-11

**Rating:** 7
**Confidence:** 4
**Ethics Flag:** 1

**Summary:**

The paper presents SlimMoE, which is a method for compressing mixture-of-experts LLMs. It advocates for iterative pruning and distillation, first slimming the experts (without removing entire experts) and attention heads, then distilling from the original LLM, and repeated 2-3 times.   The paper applies this method to Phi 3.5-MoE and derives two models, Phi-mini-MoE and Phi-tiny-MoE, which have 18% and 9% of the original total parameters, respectively. The paper compared the resulting models with other recent MoE and dense models of similar sizes, and showed their compressed models outperformed the other baselines that have similar active parameters. In additional ablations, the paper demonstrates that their multi-stage approach outperforms single-stage prune-and-distill, the dense version of Phi is less amenable to aggressive pruning than MoE, longer initialization stages is beneficial for model quality, and that their pruning method outperforms Seer-MoE and M-SMoE at aggressive compression ratios.

**Questions To Authors:**

(Q1) Does multi-stage distillation outperform one-stage distillation and are MoE models easier to prune for any other model family than Phi/Phi-MoE? I believe additional evidence OR discussion on the limitations of the existing evidence would help readers understand the take-aways of Sec 4.2 and 4.3.

(Q2) Would multi-stage pruning/distillation still outperform single-stage at lower token budgets (E.g. <100B tokens)? Additional discussions would help improve the paper.

**Reasons To Accept:**

1. The paper presented substantial experiments conducted at large scales (~400B token distillation), which in itself is challenging to execute and I believe is a valuable data point for the community and future research.
2. The final compressed models are quite strong compared to other MoE models in similar size classes, indicating that compression from a larger MoE may be the preferred method to obtain smaller high-quality MoEs.
3. Ablation experiments explaining the design choices are helpful, I especially appreciated the result in Figure 2.

**Reasons To Reject:**

(W1) The method is only evaluated on the Phi-MoE family of models. Although the final results are strong, the paper draws general conclusions from them in Sec 4.2 "Multi-stage vs. OneStage" and Sec 4.3 "Are MoE Models Easier to Prune?". As the authors themselves stated in Sec 4.4.2, some of their results differ from prior works, which they primarily attributed to differences between model families. This calls into question if the conclusions from Sec 4.2 and 4.3 are specific to Phi-MoE or can be applied to other model families.

(W2) Distillation at ~400B tokens seems quite resource intensive even though it's 10% of the original pretraining data. It is not explained whether the token horizon is a crucial factor for the method to work. In comparison, Minitron (https://arxiv.org/pdf/2407.14679) studied distillation using 90B tokens (2.5% of original pretraining data).

---

> ### Author Response · Authors · 2025-06-02
>
> Thank you for your thoughful comments! Please see our response to your concerns below:
>
> **(W1) The method is only evaluated on the Phi-MoE family of models. This calls into question if the conclusions from Sec 4.2 and 4.3 are specific to Phi-MoE or can be applied to other model families.**
>
> We appreciate this important concern about the generalizability of our conclusions. We remark that our multi-stage approach is architecturally agnostic and contains no Phi-specific design choices. To address the generalizability of our multi-stage approach (Section 4.2), we are currently conducting experiments on Mixtral 8x7B to provide additional evidence beyond the Phi-MoE family. Given the computational requirements of these large-scale experiments, we will do our best to provide these results before the discussion phase ends.
>
> Regarding Section 4.3's conclusion about MoE models being easier to prune, we believe using Phi series leads to more reliable conclusions than using other model families. Phi-medium and Phi-MoE use the same pre-training data and similar training settings. Both of them are trained from-scratched and well-converged, and achieve similar performance in the end. In contrast, other model families do not disclose the differences in their MoE and dense training and often have large performance gaps. Hence, using them for experiment may not be able to get a fair comparison and draw meaningful conclusion. In our revision, we will present these findings as well-supported conclusions within the scope of our experimental evidence while noting the need for broader validation across diverse architectures.
>
> **(W2, Q2) Distillation at ~400B tokens seems quite resource intensive even though it's 10% of the original pretraining data. It is not explained whether the token horizon is a crucial factor for the method to work. In comparison, Minitron studied distillation using 90B tokens. Would multi-stage pruning/distillation still outperform single-stage at lower token budgets (E.g. <100B tokens)?**
>
> Thank you for raising this practical concern. Our method remains effective with significantly reduced token budgets. To demonstrate this, we evaluated our approach using only 90B total tokens (with 25B for the first stage) on Phi-mini-MoE:
> | Method      | MMLU  | Arc-C | Winograde | Hellaswag |
> | ----------- | ----- | ----- | --------- | --------- |
> | One-stage   | 64.11 | 59.13 | 72.34     | 73.23     |
> | Multi-stage | 65.59 | 62.54 | 72.74     | 73.38     |
>
> We can see that our multi-stage approach notably outperforms the one-stage baseline, demonstrating the method's effectiveness across different training horizons.
>
> We use a larger token budget than Minitron mainly due to our significantly more aggressive compression ratios: 42B→3.8B (9% retention) and 42B→7.6B (18% retention) with 400B tokens, compared to Minitron's 15B→8B (53% retention)/8B→4B (50% retention) with 90B total tokens. Higher compression ratio often requires a larger token budget for recovery to maintain performance. Furthermore, practitioners can adjust the training horizon based on available resources, though with some performance trade-offs.
>
> **(Q1) Does multi-stage distillation outperform one-stage distillation and are MoE models easier to prune for any other model family than Phi/Phi-MoE? I believe additional evidence OR discussion on the limitations of the existing evidence would help readers understand the take-aways of Sec 4.2 and 4.3.**
>
> We are working to provide Mixtral experiments to strengthen the evidence for our multi-stage approach's general applicability.
> For the conclusion that MoE models are easier to prune (Section 4.3), we believe this finding has broader validity because our experimental design controls for most confounding factors—both Phi-medium and Phi-MoE share the same pretraining data and similar procedures, isolating the architectural differences. Using other model families for experiment may not be able to get a fair comparison and draw meaningful conclusion. However, we acknowledge that general claims require more evidence. In our revision, we will provide clearer discussion regarding the generalizability.

---

> > ### Comment · Reviewer_MkJW · 2025-06-08
> >
> > I thank the authors for the detailed response and will raise my score. I suggest including the lower token budget experiments and discussions around generalizability to other model families.

---

> > > ### Author Response · Authors · 2025-06-10
> > >
> > > Thank you for raising your score and thoughtful comments! We will make sure to include the discussions. We would also like to share some updates on our experiments using Mixtral as the target model, as promised, to support the generalization of SlimMoE's multi-stage approach. We pruned the pretrained Mixtral 8x7B model to 20% on MoE layers and 50% on attention layers. Using a total training budget of 32B tokens, our multi-stage approach allocates 9.6B tokens to first-stage initialization. As presented below, the results show consistent gains across tasks, confirming that our multi-stage approach can generalize to other model families!
> > >
> > > | Method      | MMLU  | Arc-C | Winogrande | HellaSwag |
> > > | ----------- | ----- | ----- | ---------- | --------- |
> > > | One-stage   | 49.11 | 46.98 | 65.06      | 59.61     |
> > > | Multi-stage (SlimMoE) | **52.71** | **50.85** | **68.98**      | **63.30**     |
> > >
> > > Thank you again for your valuable feedback throughout this review process.

---

### Official Review · Reviewer_e3xh · 2025-05-12

**Rating:** 6
**Confidence:** 3
**Ethics Flag:** 1

**Summary:**

This paper presents SlimMoE, a method that systematically reduces parameter counts by slimming experts. The authors claim the proposed method effectively mitigate the performance degradation in one-shot pruning approaches. The authors applied the approach to Phi-3.5-MoE and creates two variants using 400B tokens, and then evaluated the downstream performance of these variants.

**Questions To Authors:**

see above ^^. I am happy to adjust my score if the authors could clarify a bit more.

**Reasons To Accept:**

- The topic is very timely and relevant.
- The proposed approach is able to significantly reduce the parameter count of MoE models. It is a systematical approach that might be interesting to the COLM community as well as MLSys community.
- The authors evaluate the model performance with both MoE and dense models of similar sizes, across multiple downstream benchmarks. The evaluation demonstrates the effectiveness of the proposed approach.

**Reasons To Reject:**

- The authors mentioned "Our experiments demonstrate that these compressed models outperform others of similar size and remain competitive with larger models. For instance, Phi-mini-MoE achieves similar MMLU scores to Llama 3.1 8B despite having significantly lower latency.". However, the latency figure is only shown in the Appendix Figure 4, which I believe is not mandatory to read. It would be a lot better to me if this figure is promoted to the main body to show a full picture of the serving performance vs. the model accuracy. Not a reason for rejection but I think can significantly improve the paper presentation.
- I also have a few questions/concerns regarding this figure, together with the accuracy table (Table 2).

    - It is true that "Phi-mini-MoE achieves similar MMLU scores to Llama-3.1 8B with lower latency", but also I read that "Phi-mini-MoE achieved 5% lower on MMLU compared to Phi-3-small, which has the same number of parameters with Phi-mini-MoE but just activated all of them", I am thus wondering about the latency benefit of Phi-mini-MoE, but I cannot find the performance of Phi-3-Small in Figure 4. What is the rationale behind selecting these particular models in Figure 4? Could the authors elaborate the performance of these two models?
    - I failed to find the rationale behind comparing to Llama-3 series. Ideally I would like to see a comparison where: 1) you have exactly the same training data 2) the only difference is the model architecture (MoE vs. Non-MoE) and the number of parameters. 3) then you compress the MoE version and compare the performance with the dense version but with similar parameter count. While I understand the perfect/ideal comparison is hard to achieve, I don't think I fully understand why the authors pick Llama-3 series model as a baseline in the abstract. Are they using the same training data? Otherwise isn't Phi-3 series a better baseline and worth highlight in the abstract? Across the paper I saw authors use Phi-3 series a lot but I think this is also worth strengthening in the abstract.

- 10% of the training data (400B tokens) sounds a lot to me since this is the pre-training stage. I am not sure how practical the proposed approach is when the model gets larger, as training a large model for few hundred billion tokens sounds like an expensive process. This is a minor point though and should not be a show-stopper.
- Related to the above question, the authors only showed the experiments on Phi- series model only. Would the proposed method generalize to other model architectures? Maybe I missed something but why the authors specifically choose this series of models? [not a reason to reject but genuinely curious]
- How did the authors choose these 400B tokens in the dataset? Is it uniformly selected from the Phi report? Since you have used 400 billion tokens in addition to the standard Phi-3 series, would a continued pre-trained Phi-3 series on the same 400 billion tokens form a stronger baseline? It sounds a little bit unfair to me since you have trained on extra 400B tokens.

---

> ### Author Response · Authors · 2025-06-02
>
> **5. The authors only showed the experiments on Phi- series model only. Would the proposed method generalize to other model architectures?**
>
> Our method is architecturally agnostic and contains no Phi-specific design choices. To better support the generalization ability of the proposed method, we are conducting experiments on Mixtral 8x7B models. Due to the large size and computational resources needed. We will try our best to provide the results before the discussion phase ends.
> We choose Phi-3.5-MoE for main experiments for several reasons:
> 1. Phi-3.5-MoE (42B parameters) represents the largest MoE model we could comprehensively study given our computational resources, while still being substantial enough to demonstrate meaningful compression benefits. Larger models like DeepSeek-V3 exceed our current experimental capacity.
> 2. Phi-3.5-MoE achieves strong benchmark performance, which may be more sensitive to pruning. This can better show the difference between compression techniques.
>
>
>
> **6. How did the authors choose these 400B tokens in the dataset? Would a continued pre-trained Phi-3 series on the same 400 billion tokens form a stronger baseline?**
>
> We randomly sampled 400B tokens from the same pre-training dataset used for Phi-3 series. We also observed that continued pre-training the baseline on these same repeated tokens would not improve performance.
> As we conduct large-ratio pruning on the models, we believe using additional tokens for recovery is reasonable and is much cheaper than training a smaller model from scratch.

---

> ### Author Response · Authors · 2025-06-02
>
> Thank you for the valuable feedback! Please see our response per each of your concerns below:
>
> **1. The authors mentioned the compressed models remain competitive with larger models while achieving lower latency. But the latency figure is only shown in the Appendix. It would be a lot better to me if this figure is promoted to the main body to show a full picture of the serving performance vs. the model accuracy.**
>
> Thank you for this valuable suggestion! We agree that including the latency analysis in the main paper would provide a more comprehensive view of our models' performance-efficiency trade-offs.We have moved Figure 4 to Section 4.5 of the main paper to improve readability.
>
>
> **2. Latency performance of Phi-3-Small is not found in Figure 4. What is the rationale behind selecting these particular models? Could the authors elaborate the performance of Phi-3-small and Phi-mini-MoE?**
>
> In figure 4, we select models with similar performance levels (Llama 3.1 8B, Phi 3 mini) or MoE models with similar activated parameter counts (Qwen 1.5 MoE, DeepSeek V2 Lite) with Phi-mini-MoE. We aim to show that our model is more efficient compared to models with similar performance, and that it maintains efficiency advantages over other MoEs with similar activated parameter counts.
>
> To address your concern, We added latency profiling on Phi-3-small in the plot (results available at <https://anonymous.4open.science/r/SlimMoE-rebuttal-8F3F/latency_phismall.pdf>). The results confirm that Phi-mini-MoE (2.4B activated, 7.6B total) achieves substantially lower latency than Phi-3-small (7.4B total), demonstrating the efficiency benefits of sparse activation when compared to dense models of similar total parameter count.
>
>
>
> **3. I failed to find the rationale behind comparing to Llama-3 series. Why the authors pick Llama-3 series model as a baseline in the abstract. Isn't Phi-3 series a better baseline and worth highlight in the abstract?**
>
> Thanks for pointing this out! You are absolutely correct that Phi-3 series represents a better baseline as our comparison between Phi-3 series models is fair and satisfies the criteria – 1) same training data for the MoE and dense models, 2) only architectural difference between the MoE and dense. For 3), our results show that Phi-mini-MoE (7.6B total/2.4B activated) achieve similar or better performance compared to Phi-3-mini (3.8B) with ⅔ activated parameters. Phi-tiny-MoE (3.8B total/1.1B activated) keeps the same total parameters as Phi-3-mini while activating ⅓ of them, demonstrating significant efficiency gains with controlled performance trade-offs.
>
> We compare to the Llama-3 series in the abstract mainly to provide readers with a reference point. As Llama3 is one of the most popular and widely-recognized open-source models, comparing with it can give the reader a clear sense of the relative position of our developed models. We acknowledge that this distinction could be clearer. In the revision, we will emphasize the Phi-3 series as our primary technical baseline in the abstract, and strengthen the discussion of Phi-3 comparisons throughout the paper.
>
> **4. I am not sure how practical the proposed approach is when the model gets larger, as training a large model for few hundred billion tokens sounds like an expensive process.**
>
> We appreciate this practical concern about computational costs. While 400B tokens represents a substantial amount, we'd like to highlight several factors that make our approach more efficient even when dealing with large models:
> 1. Our multi-stage framework allocates most training tokens to the smallest target model size. Since the longest training stage operates on the heavily pruned model (E.g., Phi-tiny-MoE has 3.8B as the target size, which is significantly smaller than the original 42B parameters).
> 2. Training the target size models from scratch would require the full ~4T tokens. Using only 400B tokens (10% of original training data) to achieve competitive performance via compression represents a substantial efficiency gain.
> 3. We can pre-compute and cache teacher model logits as a one-time operation, significantly reducing memory requirements during distillation stages.
> 4. Our method allows users to adjust the token budget based on available resources. To demonstrate this flexibility, we evaluated a reduced budget variant using only 90B total tokens (25B for the first stage) on Phi-mini-MoE:
> | Method      | MMLU  | Arc-C | Winograde | Hellaswag |
> | ----------- | ----- | ----- | --------- | --------- |
> | One-stage   | 64.11 | 59.13 | 72.34     | 73.23     |
> | Multi-stage | 65.59 | 62.54 | 72.74     | 73.38     |
>
> Even with this significantly reduced budget, our multi-stage approach notably outperforms the one-stage baseline, demonstrating the method's effectiveness across different training horizon.

---

> ### Comment · Reviewer_e3xh · 2025-06-05
>
> Thank you for a detailed explanation. That clear things up and I would like to raise my score.

---

> > ### Author Response · Authors · 2025-06-10
> >
> > Thank you for raising your score and thoughtful comments! We would like to share some updates on our experiments using Mixtral as the target model, as promised, to support the generalization of SlimMoE's multi-stage approach. We pruned the pretrained Mixtral 8x7B model to 20% on MoE layers and 50% on attention layers. Using a total training budget of 32B tokens, our multi-stage approach allocates 9.6B tokens to first-stage initialization. As presented below, the results show consistent gains across tasks, confirming that our multi-stage approach can generalize to other model families!
> >
> > | Method      | MMLU  | Arc-C | Winogrande | HellaSwag |
> > | ----------- | ----- | ----- | ---------- | --------- |
> > | One-stage   | 49.11 | 46.98 | 65.06      | 59.61     |
> > | Multi-stage (SlimMoE) | **52.71** | **50.85** | **68.98**      | **63.30**     |
> >
> > Thank you again for your valuable feedback throughout this review process.

---

### Official Review · Reviewer_6UVt · 2025-05-13

**Rating:** 7
**Confidence:** 3
**Ethics Flag:** 1

**Summary:**

This paper proposes SlimMoE, a multi-stage compression framework for large Mixture-of-Experts (MoE) language models. By combining structured pruning (specifically, intra-expert neuron slimming) with staged knowledge distillation, the method compresses a large 41.9B parameter MoE (Phi 3.5-MoE) down to 7.6B (Phi-mini-MoE) and 3.8B (Phi-tiny-MoE) while maintaining strong performance across a range of benchmarks. The authors demonstrate that SlimMoE outperforms existing MoE and dense models of comparable size and provides clear ablations and theoretical motivation for each component of the framework.

**Questions To Authors:**

Could you please provide a detailed description of the 400B token subset used for distillation (e.g., domain mix, sampling method, overlap with eval sets).

**Reasons To Accept:**

- Addresses a highly practical problem — the inference and fine-tuning costs of large MoE models.

- The idea of pruning within experts (rather than removing whole experts) is effective in preserving functionality.

- A clean multi-stage prune-distill pipeline that is computationally more efficient than iterative pruning and more robust than one-shot pruning.

- Phi-mini-MoE and Phi-tiny-MoE outperform similarly-sized models and rival larger dense baselines across a wide range of tasks.

- Extensive experimental details, including intermediate architectures, training configs, and evaluation settings.

**Reasons To Reject:**

- The paper is primarily empirical. While effective, the SlimMoE method lacks a theoretical framework to explain why staged slimming and intra-expert pruning outperform other approaches (e.g., optimization landscape analysis, convergence guarantees).

- Although well-executed, the framework combines known techniques (pruning and distillation) rather than introducing fundamentally new ideas. A deeper theoretical analysis. For example, explaining why expert slimming better preserves model capacity than expert dropping, would strengthen the conceptual contribution.

- While SlimMoE reduces inference costs, the training process still requires substantial compute (64 A100 GPUs). Additional discussion on low-resource training setups or lighter teacher models would improve accessibility.

- The paper does not compare SlimMoE to quantization or LoRA-based compression techniques. While it benchmarks against pruning-based and dense-model baselines, other popular compression strategies such as quantization-aware training (QAT), Low-Rank Adaptation (LoRA), or weight sharing are not considered.

Minor Issues
- Missing period in the middle of line 206.
- Inconsistent terminology: unify the usage of “onestage”/“one-stage” (e.g., lines 236, 250) and “multi-stage”/“multistage” (e.g., line 328).
- Missing publication year for the reference of He et al.
- Move latency comparisons and efficiency plots into the main body of the paper for better visibility.
- Include a system diagram to illustrate the SlimMoE multi-stage prune-and-distill workflow.

---

> ### Author Response · Authors · 2025-06-02
>
> Thank you for your positive review! Please see our response to each of your concerns below:
>
> **1. The paper is primarily empirical. While effective, the SlimMoE method lacks a theoretical framework to explain why staged slimming and intra-expert pruning outperform other approaches (e.g., optimization landscape analysis, convergence guarantees).**
>
> We agree that SlimMoE is primarily empirical. However, this approach aligns with the current state of MoE compression research, where theoretical analysis remains exceptionally challenging due to the complex routing dynamics and deep architectures involved. Recent MoE compression methods such as Seer-MoE [1] and [2] are similarly empirical and do not offer convergence guarantees or formal optimization analyses. Even in the broader context of dense LLM compression, established works like LLM-Pruner [4] and Minitron [5] rely primarily on empirical validation rather than theoretical guarantees.
> The complexity of analyzing MoE models stems from their non-trivial routing mechanisms and the interactions between multiple experts across deep layers. To our knowledge, only one recent work [3] provides theoretical guarantees for MoE expert pruning, but their analysis is limited to simplified single-layer MoE architectures on binary classification tasks, which does not extend to the deep MoE LLMs we studied. Given this landscape, our contribution lies in providing comprehensive empirical analysis and demonstrating the practical effectiveness of SlimMoE across multiple benchmarks and compression ratios. We will include this discussion of the theoretical challenges inherent to MoE analysis in our revision.
>
> ### Reference
> [1] Seer-moe: Sparse expert efficiency through regularization for mixture-of-experts
>
> [2] Not all experts are equal: Efficient expert pruning and skipping for mixture-of-experts large language models.
>
> [3] A Provably Effective Method for Pruning Experts in Fine-tuned Sparse Mixture-of-Experts
>
> [4] LLM-Pruner: On the Structural Pruning of Large Language Models
>
> [5] Compact language models via pruning and knowledge distillation.
>
>
> **2. Although well-executed, the framework combines known techniques (pruning and distillation) rather than introducing fundamentally new ideas. A deeper theoretical analysis. For example, explaining why expert slimming better preserves model capacity than expert dropping, would strengthen the conceptual contribution.**
>
> While SlimMoE builds upon established pruning and distillation techniques, our systematic integration of multi-stage design with intra-expert slimming represents a novel approach specifically designed for MoE architectures. We can provide more insight into why expert slimming outperforms expert dropping through sensitivity analysis. Using Taylor expansion, when parameters change by Δw, the loss function expands as:
> $$
> \mathcal{L}(\mathbf{w} + \Delta \mathbf{w}) \approx \mathcal{L}(\mathbf{w}) + \nabla \mathcal{L}(\mathbf{w})^\top \Delta \mathbf{w} + \frac{1}{2} \Delta \mathbf{w}^\top H(\mathbf{w}) \Delta \mathbf{w}
> $$
> For pruning (setting $\Delta w_i = -w_i$), the loss increase becomes:
> $$
> \Delta \mathcal{L} \approx - \sum_i \frac{\partial \mathcal{L}}{\partial w_i} w_i + \frac{1}{2} \sum_i H_{ii} w_i^2
> $$
> In our experiment, we use the sensitivity scores $|\frac{\partial \mathcal{L}}{\partial w_i} w_i|$ as pruning criterion and observe that scores are similarly distributed across experts, indicating that representational knowledge is distributed across all experts rather than concentrated in few experts. Expert dropping removes all parameters of an expert simultaneously, potentially eliminating unique sub-functions critical for certain token distributions, resulting in large cumulative $\Delta \mathcal{L}$. In contrast, expert slimming selectively removes only the least sensitive weights within each expert, preserving high-impact neurons and maintaining expert specialization. This provides a more principled, fine-grained approach that better maintains model capacity through smaller overall loss changes.
> As no existing rigorous theoretical framework fully captures how deep MoE models retain knowledge and how pruning affects, our analysis provides intuitive and empirically-supported insights that advance understanding in this domain.

---

> > ### Author Response · Authors · 2025-06-02
> >
> > **3. While SlimMoE reduces inference costs, the training process still requires substantial compute (64 A100 GPUs). Additional discussion on low-resource training setups or lighter teacher models would improve accessibility.**
> >
> > We agree that the 64 A100 GPU requirement represents a substantial computational investment. We provide more discussion on the computation cost as follows: In our experiments, we used a 400B token budget as we found our models converge at this level in high-compression scenarios. To examine our method's performance in lower-computation setups, we evaluated our approach using only 90B total tokens (with 25B for the first stage) on Phi-mini-MoE:
> > | Method      | MMLU  | Arc-C | Winograde | Hellaswag |
> > | ----------- | ----- | ----- | --------- | --------- |
> > | One-stage   | 64.11 | 59.13 | 72.34     | 73.23     |
> > | Multi-stage | 65.59 | 62.54 | 72.74     | 73.38     |
> >
> >
> > We can see that our approach notably outperforms the naive baseline, demonstrating the method's effectiveness on shorter training horizons. This allows practitioners to adjust the computational budget based on available resources while maintaining the core advantages of our approach.
> >
> > Several design choices in our framework also improve computational efficiency. Our multi-stage design allocates most training tokens to the smallest target model size, meaning the longest training stage operates on the heavily pruned model rather than the full 42B parameter teacher. This creates favorable computational scaling, particularly when targeting aggressive compression ratios. Furthermore, the alternative of training compressed models from scratch would require approximately 10× more computation to achieve competitive performance, making our distillation approach substantially more efficient. Additionally, we can pre-compute and cache teacher model logits as a one-time operation, significantly reducing memory requirements during the distillation stages and enabling more efficient resource utilization throughout the training process.
> >
> > **4. The paper does not compare SlimMoE to quantization or LoRA-based compression techniques. While it benchmarks against pruning-based and dense-model baselines, other popular compression strategies such as quantization-aware training (QAT), Low-Rank Adaptation (LoRA), or weight sharing are not considered.**
> >
> >
> > Thank you for highlighting other compression strategies. While quantization, LoRA, and weight sharing represent important compression approaches, they address fundamentally different aspects of model efficiency compared to SlimMoE's structural pruning approach. Quantization methods like MoQE [6,7] reduce memory through lower numerical precision rather than architectural changes, requiring specialized hardware for acceleration and having inherent lower bounds on compression ratios. LoRA-based approaches [8] cannot achieve the aggressive compression ratios we demonstrate because MoE experts typically lack the low-rank structure necessary for effective factorization. Weight sharing techniques [9] reduce parameters by forcing experts to share weights, but this constrains expert specialization and limits the model's representational capacity compared to our selective pruning approach.
> >
> > Importantly, these compression strategies operate on different dimensions and are complementary rather than competitive with SlimMoE. Our method operates at the architectural level by removing structural redundancy through expert pruning and distillation, while quantization, LoRA, and weight sharing compress model weights without altering expert topology. A SlimMoE-compressed model can be further enhanced with these techniques for additional memory savings and efficiency. SlimMoE attacks structural overcapacity while other methods target representational or precision redundancy, enabling these approaches to be composed for even deeper compression. We will add discussion of these compression methods to our revision.
> >
> >
> > ### Reference
> > [6] Mixture of Quantized Experts (MoQE): Complementary Effect of Low-bit Quantization and Robustness
> >
> > [7] MoEQuant: Enhancing Quantization for Mixture-of-Experts Large Language Models via Expert-Balanced Sampling and Affinity Guidance
> >
> > [8] MoE-I2: Compressing Mixture of Experts Models through Inter-Expert Pruning and Intra-Expert Low-Rank Decomposition
> >
> > [9] Delta Decompression for MoE-based LLMs Compression
> >
> > **Minor Issues**
> > Thanks for your detailed review! We will make sure to correct the errors and improve the readability in our revision.

---

> > > ### Comment · Reviewer_6UVt · 2025-06-05
> > >
> > > Thank you for a detailed explanation. After going through the response, I would like to keep my already positive score as it is.

---

### Decision · Program_Chairs · 2025-07-08

**Decision:**

Accept

**Comment:**

In this paper the authors propose a methodology for compressing MoE language models that combines structured pruning (within experts) with knowledge distillation (general-purpose, on hundreds of billions of tokens of pretraining data). They demonstrate the approach on Phi-3 MoE models (1-7B range) across a variety of tasks (e.g. MMLU, GSM8k, arc-c, human-eval) and demonstrate that the approach achieves comparable end-task performance to existing models while using 30% of the active parameters. They also present some experiments suggesting that MoE models are more compressible than dense models.

Reviewers agreed that the proposed approach represents a well-motivated solution to a practical issue, and that the paper presented strong empirical results demonstrating the efficacy of the approach towards this end. Reviewers shared concerns about the generalizability of the approach beyond the Phi-3 family of models, and regarding the computational burden of the distillation component, which requires fine-tuning on hundreds of billions of tokens of pretraining data. The authors provided some preliminary additional experiments on Mixtral models and smaller budgets of training tokens for the distillation phase during the discussion period, and if the paper is accepted the authors should include the full set of results (evaluation tasks) for those settings in order to address those weaknesses.